Baseline haematological parameters in three common Australian frog species

Jadwani-Bungar Tara
Doidge Nicholas P.
Wallace Danielle K.
http://orcid.org/0000-0003-2975-9494 Brannelly Laura A. laura.brannelly@unimelb.edu.au
Melbourne Veterinary School, University of Melbourne , Werribee, VIC , Australia
Gottdenker Nicole
Electronic publication date: 2024 Jun 7
Publication date: 2024
Volume: 12
Electronic Location ID: e17406
Received 2023 Sep 13; Accepted 2024 Apr 25
Copyright: © 2024 Jadwani-Bungar et al.
Copyright year: 2024
Copyright holder: Jadwani-Bungar et al.
License: This is an open access article distributed under the terms of the Creative Commons Attribution License, which permits unrestricted use, distribution, reproduction and adaptation in any medium and for any purpose provided that it is properly attributed. For attribution, the original author(s), title, publication source (PeerJ) and either DOI or URL of the article must be cited.
License URL: https://creativecommons.org/licenses/by/4.0/

Keywords: Differential cell counts, Haematological parameters, Frog, Blood count, Leukocyte, Lymphocyte

Funding: Australian Research Council DE180101395 The University of Melbourne The project was funded by the Australian Research Council (DE180101395) and the University of Melbourne. The funders had no role in study design, data collection and analysis, decision to publish, or preparation of the manuscript.

==============================
Amphibians are experiencing declines globally, with emerging infectious diseases as one of the main causes. Haematological parameters present a useful method for determining the health status of animals and the effects of particular diseases, but the interpretation of differential cell counts relies on knowing the normal ranges for the species and factors that can affect these counts. However, there is very little data on either normal haematological parameters or guides for blood cell types for free-ranging frog species across the world. This study aims to 1) create a visual guide for three different Australian frog species: Litoria paraewingi, Limnodynastes dumerilii, and Crinia signifera, 2) determine the proportions of erythrocytes to leukocytes and 3) differential leukocytes within blood smears from these three species and 4) assess the association between parasites and differential counts. We collected blood samples from free-ranging frogs and analysed blood smears. We also looked for ectoparasites and tested for the fungal disease chytridiomycosis. Overall, we found that the differentials of erythrocytes to leukocytes were not affected by species, but the proportions of different leukocytes did vary across species. For example, while lymphocytes were the most common type of leukocyte across the three species, eosinophils were relatively common in Limnodynastes dumerilii but rarely present in the other two species. We noted chytridiomycosis infection as well as ectoparasites present in some individuals but found no effect of parasites on blood parameters. Our results add baseline haematological parameters for three Australian frog species and provide an example of how different frog species can vary in their differential blood cell counts. More information is needed on frog haematological data before these parameters can be used to determine the health status of wild or captive frogs.

Introduction

Amphibians are currently the most threatened vertebrate class (Howard & Bickford, 2014). Over the past 30 years infectious diseases have emerged as one of the main causes of their decline, resulting in the development of different approaches to amphibian conservation (Davis, Maney & Maerz, 2008, Forzán et al., 2017). Haematological parameters, including cytology, present useful methods for determining the health status of an animal and the effects of a particular disease on their immune system. The immune system of animals is comprised of several components which interact in complex but predictable fashions to mediate immune responses (Day & Schultz, 2014). Leukocytes are one of these essential components. The classification of leukocytes is based on cell morphology in stained blood smears, as well as cytochemical and immunocytochemical characterisation (Bricker, Raskin & Densmore, 2012). Frog leukocytes have been characterised as lymphocytes, monocytes, neutrophils, basophils, and eosinophils (Davis & Durso, 2009; Forzán et al., 2017).

Leukocyte differentials are particularly useful because they change in response to stress or pathology (Davis, Maney & Maerz, 2008) and can therefore assist with diagnoses. These changes are highly conserved across vertebrate taxa (Davis, Maney & Maerz, 2008). However, for most free-ranging amphibian species we do not have the reference intervals necessary for accurate analysis of leukocytes (Davis, Maney & Maerz, 2008, Forzán et al., 2017). In many amphibians there are not even visual guides for what the different cell types look like.

Leukocytes among different frog species show morphological differences which pose a challenge when performing cytology. A level of experience and expertise is required to correctly identify different types of cells, and there is high variability in the leukocyte profiles of different species, including between those within the same genus (Forzán et al., 2017). Furthermore, there can be large variation and variability in poikilotherms based on many intrinsic and extrinsic factors (Ahmed, Reshi & Fazio, 2020). Consequently, it is difficult to establish reference intervals and make comparisons among different species and to assess their health. Without accurate reference intervals with corresponding data on intrinsic and extrinsic factors, our ability to draw conclusions based on leukocyte profiles and disease status is limited.

In this study we conducted a field survey to sample three common Australian frog species: Litoria paraewingi, Limnodynastes dumerilii, and Crinia signifera. These three species co-occur and represent three of the five frog families that are native to Australia: Hylidae, Limnodynastidae, and Myobatrachidae (respectively). The species distribution of Lit. paraewingi is limited to the central eastern Victoria border with New South Wales, while both Lim. dumerilii and C. signifera have broad distributions along the southeast of Australia from Tasmania into Queensland and west into South Australia. These three species are not under threat of extinction and are species of “least concern”. In this study we aimed to 1) identify the different leukocyte cell types and create a visual guide, 2) determine the proportions of erythrocytes to leukocytes within blood smears from the three species, and 3) determine leukocyte differential counts in the blood smears of field-collected individuals. Finally, 4) because parasites and disease are important drivers of amphibian declines around the world, we assessed the association between parasites and differential counts. Batrachochytrium dendrobatidis is a fungal pathogen that infects frog skin and causes the devastating disease chytridiomycosis. B. dendrobatidis has been implicated in the decline of over 500 frog species globally (Scheele et al., 2019) and is present throughout Victoria, Australia (Murray et al., 2010; Skerratt et al., 2016). We collected parasite data (ectoparasite presence and B. dendrobatidis infection) for these three species to identify if parasites affected cell differential counts. The results of this study provide important baseline knowledge of the haematological parameters for three common frog species that can be adapted for use in other species across the world.

Methods

Field collection

Frogs were collected through hand-capture during nocturnal surveys (May 9th–13th, 2022) in the Strathbogie Ranges, Victoria, Australia (36°48′15″S 145°46′03E) (n = 126, see Table 1). Frogs were identified to be present at the site level by call, and once captured were identified to the species level by visual examination. Determination of sex was made using physical features such as nuptial pads and throat colour to identify adult males. Juveniles were identified by their small size and lack of male secondary sexual characteristics. Animals were weighed using spring scales (±0.1 g), and snout-to-vent length was measured using dial callipers (±0.1mm). Any ectoparasitic fly larvae that were visible were removed and preserved in 100% methanol and sent to our collaborators (J Keem at University of Melbourne, D De Angelis at La Trobe University, N Johnston at University of Technology Sydney, and K Newman at University of Melbourne) who are compiling instances of ectoparasitic fly larvae found to be causing myiasis in frogs in Australia.

Table 1 A summary of all the frogs captured and their morphometric data, separated by species and sex.

Species	Sex	Mass (g)	SVL (mm)	n	
Limnodynastes dumerilii				
	Female	28.4 ± 12.4	62.8 ± 11.6	20	
	Male	22.9 ± 9.6	52.2 ± 9.6	11	
	Juvenile	14.6 ± 13.6	52.8 ± 15.3	8	
Litoria paraewingi				
	Female	2.9 ± 1.4	36.1 ± 10.2	25	
	Male	1.8 ± 0.5	31.8 ± 3.5	13	
	Juvenile	0.7 ± 0.2	24.2 ± 2.0	17	
Crinia signifera				
	Female	1.6 ± 0.4	27.4 ± 1.7	8	
	Male	1.2 ± 0.3	24.3 ± 2.6	22	
	Juvenile	0.7	21.2	1	
Note:

Size data is presented as mean mass in g and mean snout to vent length (SVL) in mm, with standard deviation. N is the number of animals collected.

All frogs were swabbed for B. dendrobatidis following standard methods (Boyle et al., 2004, as stated below). Blood (approximately 5 μL per frog) was then collected from the facial (maxillary) vein of each frog according to Forzán et al. (2012), and a blood smear was made. Frogs were then released at their original collection site on the same night.

Batrachochytrium dendrobatidis swabbing and qPCR

A dry sterile rayon swab (MW-113, Medical Wire and Equipment, Wiltshire, UK) was rolled five times along each of: the ventrolateral surface of the abdomen, the ventral surface of all limbs and digits, and the medial surface of the thighs. The tip of the swab was then broken off into a 1.5 mL Eppendorf tube and stored at −20 °C within 2 h of sample collection. The samples remained in −20 °C until processing (approximately 3 months). Skin swab samples were extracted using PrepMan Ultra (Thermo Fisher Scientific, Waltham, MA, USA). The extraction method followed the manufacturer’s directions which included: adding 50 μL of PrepMan Ultra and 30–40 mg of 0.5 mm silica beads (BioSpec, Bartlesville, OK, USA) to each sample, then homogenizing the samples (using a cell homogenizer) for 2 min at 1,400 oscillations per sec, incubating samples at 95 °C to lyse the cells for 10 min, and collecting and diluting the supernatant 6:100 in ultra pure water before directly analysing for pathogen presence and quantity using qPCR (Brannelly et al., 2020). The remaining extracted DNA was stored at −20 °C. With every extraction performed one B. dendrobatidis positive control sample (zoospores from culture) and one negative control (swab only) was extracted.

We used qPCR (Rotor-Gene Q, Qiagen, Hilden, Germany) to amplify and quantify the B. dendrobatidis DNA in each sample following (Boyle et al., 2004). We used a 15 μL reaction volume, with 5 μL of template DNA, 7.5 μL of Lo-ROX 2x master mix (SensiFAST, Bioline, London, UK), and a final reaction concentration of 900 nM ITS1–3 (Purification: RP-Cartridge Gold, Sequence: CCTTGATATAATACAGTGTGCCATATGTC, Eurogentec, Integrated Science, Liège, Belgium), 900 nM 5.8S Chytr (Purification: RP-Cartridge Gold, Sequence: AGCCAAGAGATCCGTTGTCAAA, Eurogenetec, Integrated Science, Liège, Belgium), 250 nM Chytr MGB2 5′ 6-FAM-labelled probe (5′ 6-FAM - CGA-GTC-GAA-CAA-AAT - MGB-Eclipse® 3′; Integrated Science, Liège, Belgium) and 400 ng/μL bovine serum albumin (Thermo Fisher Scientific, Waltham, MA, USA). The amplification conditions were initial denaturation conditions of 2 min at 50 °C, 10 min at 95 °C, followed by 40 cycles of denaturation at 95 °C for 15 s and annealing at 60 °C for 1 min. On each qPCR reaction plate we included a series of seven plasmid-based B. dendrobatidis standards (purchased from Pisces Molecular, Bolder, Colorado, US containing 4.2, 42, 420, 4,200, 42,000, 420,000, and 4,200,000 DNA copies per reaction). On each qPCR reaction plate we also included a no template control, where ultrapure water replaced the template DNA. The extraction positive and negative control samples were analysed via qPCR like all sample-extracted DNA.

DNA copies of the B. dendrobatidis ITS gene extracted from each swab were estimated by extrapolating the gene copies detected by qPCR (Rotor-Gene Q 2.3.5 software), considering the elution volume and dilution of template DNA. Each sample was determined to be B. dendrobatidis positive if its reaction well was amplified to at least two ITS DNA copies detected. If the sample was considered negative for B. dendrobatidis, the B. dendrobatidis DNA copies was coded as 0 for that sample (Brannelly et al., 2020). Samples were analysed via qPCR in singlicate, which is a common and accepted practice for chytrid infection studies (Brannelly et al., 2015, 2017). All positive extraction controls were positive, all negative extraction controls were negative, and all no template qPCR controls were negative for B. dendrobatidis DNA copies.

Blood sampling and analysis

Blood samples were taken in accordance with Forzán’s protocol for collecting blood from the facial (maxillary) vein (Forzán et al., 2012). A 20G needle was used to pierce the skin and score the vein between the upper jawline and the rostral side of the tympanum (Forzán et al., 2012). The score released blood onto the skin surface that was then collected with a heparinised capillary tube. Blood from the capillary tube was transferred onto microscope slides, smeared, and left to air dry. Immediately after samples dried (within 2 h after collection), they were fixed with 100% methanol and stained using DiffQuik, air-dried, and then mounted on a coverslip using Permamount™ mounting medium (Thermo Fisher Scientific, Waltham, MA, USA).

Blood samples were examined using light microscopy to a) visually identify the different cell types within the blood smears, b) perform a differential cell count of leukocytes and erythrocytes within the smear (i.e., the percentage of leukocytes and erythrocytes), c) perform a differential cell count of leukocytes (i.e., the percentage of the different types of leukocyte cells), and d) detect the presence of any haemoparasites using visual microscopic assessment. We performed these analyses on blood smears that were high quality: e.g., few broken cells, few blood clots, effective staining, high quality smears, large enough blood volume within the sample. Blood smears were examined in the monolayer at a total magnification of 1,000× (100× oil immersion objective lens).

While standard cytological analysis includes assessing erythrocyte concentrations using a hemocytometer (Forzán et al., 2016), we were unable to perform these analyses due to the limited volume of blood collected from each frog. Therefore, we used an adapted analysis and performed a differential count of leukocytes to erythrocytes within the smear, where all cell types were counted until at least 500 total cells were reached (including erythrocytes, thrombocytes, and leukocytes) to calculate their relative proportion within the sample. To perform a differential leukocyte count, we counted until a minimum of 100 leukocytes was reached (Chabot-Richards & George, 2015). The leukocyte differential count included the number of lymphocytes, monocytes, neutrophils, basophils, and eosinophils, which were used to calculate their relative proportion within the smear sample.

An animal was considered positive for parasites if there were macroparasites or the animals had infection with B. dendrobatidis based on the qPCR results.

Ethics and permits

The work was carried out under The University of Melbourne Animal Ethics application number 20062, which is conducted in compliance with the Australian code for the care and use of animals for scientific purposes and Wildlife Act 1975 Research Authorisation for the Victorian Department of Environment Land Water and Planning permit number 1000984. In accordance with our animal ethics application, we took all measures possible to follow the 3R tenets as described within the methods above. All fieldwork protocols conducted were compliant with The Australian Government Department of the Environment and Energy Threat Abatement Plan for the infection of amphibians with chytrid fungus resulting in chytridiomycosis (2016).

Statistical analyses

All analyses were conducted in R/RStudio using the packages ‘glmmTMB’ and ‘car’ (RStudio Team, 2020; R Core Team, 2022; Brooks et al., 2023; Fox et al., 2023). We used an alpha value of p = 0.05. Effect size was estimated using Cohen d’s statistic, and Tukey’s post hoc tests were conducted where appropriate using the package ‘emmeans’ (Lenth et al., 2022). Model assumptions were assessed to ensure no assumption was violated. For each analysis we conduced, we assessed the best fit model using the Akaike Information Criterion (AIC), a criterion used to compare statistical models to determine the best fit for data. We chose the model for each analysis conducted with the lowest AIC. If two models were <2 AIC units in difference, we chose the least complex model. For the statistical analyses that we performed, the possible predictor variables were species (Lit. paraewingi, C. signifera, and Lim. dumerilii), sex (male, female, and juvenile), size (mass, g), parasite infection status (1 = ectoparasites or B. dendrobatidis infection; 0 = no parasites observed) where appropriate, and two-way interactions among the predictor variables.

To compare differentials in the animals that had no signs of parasites (n = 79) we used generalised linear models (GLM) with a beta distribution. To assess the differential of erythrocyte to leukocyte cells, the ratio of leukocytes to erythrocytes was the response variable, and the predictor variables were species and sex. To compare the leukocyte differential counts, we conducted a series of GLMs where the proportions of the five different leukocyte cell types (lymphocyte, neutrophil, monocyte, basophil, and eosinophil) were the response variables, and species was the predictor variable.

To understand if there was an effect of parasites on the cell differentials, we investigated only the results from Lit. paraewingi and C. signifera because we did not find parasites in Lim. dumerilii (n = 70). We used generalised linear models (GLM) with a beta distribution to compare differentials across both species and sex in the animals that had no signs of parasites. To assess the differential of leukocyte to erythrocyte counts, the ratio of leukocytes to erythrocytes was the response variable, and the predictor variables were species, parasite status, and the interactive effect of species and parasite status. To compare the leukocyte differential counts, we conducted a series of GLMs where the proportions of three different leukocyte cell types (lymphocyte, neutrophil, monocyte) were the response variables, and species and parasite status were the predictor variables. We did not include basophils and eosinophils in this analysis because the counts of either type were too low for both Lit. paraewingi and C. signifera.

Results

A total of 126 animals were captured and sampled for blood analysis, B. dendrobatidis qPCR assays, and morphometric data measurements: 40 Lim. dumerilii, 51 Lit. paraewingi, and 31 C. signifera (Table 1). We analysed 96 blood smears that were high quality from 22 Lim. dumerilii (55% of this species sampled), 41 Lit. paraewingi (80%), and 29 C. signifera (94%). All the animals that were analysed appeared healthy based on physical observation.

Parasites

Eight animals tested positive for B. dendrobatidis: five Lit. paraewingi (9.8% of the total collected) and two C. signifera (6.5%). The median B. dendrobatidis load for the infected Lit. paraewingi was 51,831 DNA copies and for C. signifera was 1,500 DNA copies. Six of the eight infected animals had corresponding blood smear slides examined.

Two Lit. paraewingi had visible fly larvae ectoparasites (Fig. 1A), one of which was severely anaemic with very few mature erythrocytes (Fig. 2). This animal was removed from the study because it was not considered clinically healthy. One C. signifera had a macroparasite found in the blood smear (Fig. 1B), which has not been identified. Using microscopic techniques, we found no confirmed haemoparasites in these blood samples.

Figure 1 Ectoparasites and possible haemoparasites found on the frogs.

(A) A frog ectoparasite fly larvae found on Litoria paraewingii. (B) A possible macroparasite found on Crinia signifera. (B) Photograph taken at 20× magnification lens. Scale bar indicates 100 μm.

Figure 2 A blood smear of the Lit. paraewingii that we found with four ectoparasite fly larvae attached.

This animal was severely anaemic and clinically unwell. The blood smear shows primarily leukocytes, and there are very few mature erythrocytes in the entire smear (and only one visible in this photo–see Figs. 3A–3E for normal frogs of the same species). This was the only Lit. paraewingii animal that we found with eosinophils in the blood (arrow: with a vacuole). The fly ectoparasites and blood smear results were not included in this study. The photo was taken at 100× magnification lens, the scale bar indicates 20 μm.

Figure 3 Photographs of leukocytes found in the three study species Litoria paraewingi, Crinia signifera and Limnodynastes dumerilii.

The cell types observed are lymphocyte (L), neutrophil (N), monocyte (M), eosinophil (E), basophil (B), thrombocytes (T), immature erythrocyte (Imm), and mitotic erythrocyte (Mit). (A–E) are photos taken from Litoria paraewingii. Panel d is an eosinophil from the clinically unwell animal with ectoparasites, with a corresponding smear also shown in Fig. 2. (F–L) are photos taken from Crinia signifera. (M–R) are photos taken from Limnodynastes dumerilii. All photographs were taken using a 100x magnification oil immersion lens. The scale bar indicates 20 μm. The smears were stained with Diff-Quik stain on the night of collection once smears were dry. While the protocol was consistent for staining across species and nights, there is some inconsistency in the stain uptake, which is clear in these photographs.

In total, eight individual frogs were considered parasite positive (either had macroparasites (ectoparasitic fly larvae or unidentified macroparasite) or B. dendrobatidis fungal infection: five Lit. paraewingi (12% of the samples included in analysis; one female, one juvenile, three males) and three C. signifera (10%; three males). No individuals within this study had both B. dendrobatidis fungal infection and a coinfection with another parasite observed here. No parasites were detected in Lim. dumerilii.

Cell differentials of healthy animals

In this study we found that the median proportional ratio of cells within the blood smear was 0.946 erythrocyte cells: 0.034 leukocyte cells: 0.020 thrombocyte cells (Figs. 3 and 4). All three species had equivalent proportions of leukocyte to erythrocyte cells within their blood smears, and the proportions were equivalent in males, females, and juveniles (GLM: species, χ22 = 0.784, p = 0.676; sex, χ22 = 4.167, p = 0.125).

Figure 4 The proportion of erythrocytes to leukocytes to thrombocytes found across the animals that were clinically healthy and showed no signs of parasites.

The middle line in the box and whiskers plot indicates the median value, the upper and lower box edges indicate the first and third quartile, and the whiskers represent the 95% confidence intervals. Each point represented an individual frog.

We found an effect of species on leukocyte proportions within the blood, indicating that the proportions of different leukocyte cells varied widely across species. Species was a significant predictor for four of the five leukocyte cell types (GLM for lymphocyte proportion: χ22 = 35.343, p < 0.001; GLM for monocyte proportion: χ22 = 11.912, p = 0.002; GLM for neutrophil proportion: χ22 = 45.914, p < 0.001; GLM for eosinophil proportion: χ22 = 131.580, p < 0.001; GLM for basophil proportion: χ22 = 5.183, p = 0.075; Figs. 3B and 5).

Figure 5 The relative proportion of leukocyte cell types within each species when animals are clinically healthy and parasite free.

A minimum of 100 leukocytes were counted per individual and the relative proportion of the different cell types within each species are printed here. The proportion of the same type of leukocyte was significantly affected by species, such that lymphocyte proportion was significantly higher in Lit. paraewingi compared to both Lim. dumerilii and C. signifera (Tukey’s post hoc test p > 0.001 for both comparisons with Lit. paraewingi, p = 0.973 for Lim. dumerilii–C. signifera). The neutrophil proportion in C. signifera was significantly higher compared to the other two species (Tukey’s post hoc test p > 0.001 for both comparisons with C. signifera, p = 0.619 for Lim. dumerilii–Lit. paraewingi). The monocyte proportion was significantly lower in C. signifera compared to the other two species (Tukey’s post hoc test: Lim. dumerilii–C. signifera, p = 0.004; C. signifera – Lit. paraewingi, p = 0.20; Lim. dumerilii–Lit. paraewingi, p = 0.666). Lim. dumerilii had a significantly higher proportion of eosinophils than the other species (Tukey’s post hoc test p > 0.001 for both comparisons with Lim. dumerilii, p = 0.092 for C. signifera–Lit. paraewingi). There was no effect of species on the proportion of basophils. The middle line in the box and whiskers plot indicates the median value, the edge of the boxes indicates the first and third quartile and the whiskers represent the 95% confidence intervals. Each point represented an individual frog.

In all species we found that lymphocytes were the most prevalent leukocyte cell type and occurred in the greatest proportion in Lit. paraewingi (Figs. 4 and 5). Neutrophils and monocytes were equally common in Lit. paraewingi and Lim. dumerilii, but in C. signifera, neutrophils were much more common than in the other two frog species (Figs. 4 and 5). Eosinophils were rarely found in Lit. paraewingi—only seen once in the clinically sick animal with ectoparasites that was removed from the study (Figs. 2 and 4), while they were relatively common in Lim. dumerilii (Figs. 4 and 5).

The impacts of parasites on cell differentials

When we investigated the effect of parasite status (macroparasite or fungal infection presence) on blood smear differentials in the two species that had parasites, we found no effect of parasite status on the proportion of leukocytes (GLM: species, χ21 = 0.7349, p = 0.391; parasite status, χ21 = 0.084, p = 0.7719; species*parasite status, χ21 = 1.569, p = 0.210).

We found no effect of parasite status on leukocyte differentials when we analysed the two species that had parasites. Species was a significant predictor of the proportion of lymphocytes, monocytes and neutrophils, which coincides with the analyses conducted above (Fig. 5; GLM for lymphocyte proportion: Species, χ22 = 23.900, p < 0.001; parasite status, χ22 = 0.673, p = 0.412: GLM for monocyte proportion: Species, χ22 = 9.30, p = 0.003; parasite status, χ22 = 1.508, p = 0.219: GLM for neutrophil proportion: Species, χ22 = 38.388, p < 0.001; parasite status, χ22 = 0.541, p = 0.462).

Discussion

Our study identified trends in leukocyte profiles in three free-ranging frog species. By identifying factors that affected leukocyte differential counts and examining blood smears for abnormalities, we expanded on the limited literature on frog haematology. The trends identified can provide a baseline for blood cell differentials of healthy frogs in the wild.

Leukocyte differentials

In the healthy frogs without evidence of parasites there was no effect of species or sex on the relative proportions of leukocytes to erythrocytes within the blood sample. However, when we investigated these effects on leukocyte differentials, we found that there was a significant effect of species. Across all three species, the leukocyte present in the highest proportion was lymphocytes. Lymphocytes and neutrophils make up the majority of leukocytes in amphibians, similar to mammals and birds (Davis, Maney & Maerz, 2008). While the proportion of lymphocytes varies greatly across species, in amphibians it is frequently seen as the most common leukocyte (Davis & Durso, 2009; Young et al., 2012a; Forzán et al., 2016; Chen et al., 2022).

The relatively high proportion of eosinophils in Lim. dumerilii is of particular interest because none of the Lim. dumerilii that we sampled had detectable parasites. Eosinophils in most vertebrate animals have typically been considered as effector immune cells against extracellular, multicellular parasites like helminths (Lee et al., 2010). They have also been implicated in hypersensitivity diseases and there are several known disorders of eosinophils in mammals (Lee et al., 2010; Seimon et al., 2017; Wechsler et al., 2021). However, more recent research revealed that eosinophils may be involved in a range of homeostatic processes, suggesting a role in maintaining normal health (Wechsler et al., 2021). The high proportion of eosinophils in Lim. dumerilii might indicate highly effective innate immune protection against parasite infection or be related to a different mechanism for maintaining health.

We noted that in Lit. paraewingi, eosinophils were only found in one animal that was heavily infected with an ectoparasite (four parasitic fly larvae). Though investigating sick animals was outside the scope of this study and this was a singular finding, with further exploration it may demonstrate the links between eosinophils, maintaining normal health, and immunity against parasites. There is ongoing work exploring the health impacts of ectoparasite fly larvae on Australian frog species (J. Keem, 2022, personal communication).

The blood cell differentials of neither Lit. paraewingi nor C. signifera were affected by parasite status. Certain changes in leukocyte differentials, including monocytosis, lymphopaenia, and neutrophilia, are highly conserved across vertebrate species and associated with inflammatory disease processes (Davis, Maney & Maerz, 2008; Grzelak et al., 2017). The lack of significant changes in leukocyte differentials in this study could be because we combined macroparasites and fungal infection, and these parasites might elicit different haematological responses; infection with B. dendrobatidis can affect blood differentials, and the effect is often species dependent (Young et al., 2012b; Greenspan et al., 2017). However, parasites might cause a different response to a B. dendrobatidis skin infection, therefore leaving this study design unable to ascertain small differences in the leukocyte differentials. Future studies should collect a larger sample size in order to consider the pathogens/parasites separately. More likely, we might not have an observed change in leukocyte differentials because our sample of parasite positive animals was small (only 8% of all samples analysed). Future studies exploring the health of free-ranging amphibians are needed to understand the subclinical impacts of disease.

We collected approximately 5 μL of blood from each animal; a sufficient volume to create a blood smear but insufficient for haemocytometry. Typically, absolute erythrocyte counts are performed using a haemocytometer to calculate concentration (Forzán et al., 2016). Using a blood smear to analyse a ratio of erythrocytes within the blood, we were unable to determine absolute blood counts or estimate blood cell concentrations, as these values cannot be accurately extrapolated from blood smears (Forzán et al., 2017). Future studies would benefit from using haemocytometry. Furthermore, because the entire sample was used in the smear, we were unable to perform molecular analysis on the samples to identify any haemoparasites that might have been present but missed in our cytological analysis.

Conclusion

Our results established: 1) a photographic guide for the leukocytes present in peripheral blood, 2) a differential of erythrocytes to leukocyte proportions present in the blood, 3) leukocyte differentials, and 4) possible parasites and their effect on blood differentials in three common Australian frog species. We found that there are clear species differences in the visual appearance of blood cells, and each species has a unique leukocyte profile. While there does not appear to be an effect of parasites on amphibian blood differentials, the sample size in this study was small, and more work needs to be done to explore this relationship and identify parasites in free-ranging frogs. The current baseline knowledge on clinically healthy free-ranging frogs is severely lacking, and this study demonstrates that there are broad species differences in leukocyte differentials. For example, it was common to find a large proportion of eosinophils in one species (Lim. dumerilii) while eosinophils were exceedingly uncommon in a sympatric species (Lit. paraewingi) and might only be present in clinically unwell animals. This study adds to the current understanding of health in wild frogs, although more baseline data is required to understand the pathogenesis of certain parasites and perform species comparisons.

Supplemental Information

Supplemental Information 1 Data collected and used in this study.

The data on white cell to red cell proportions; the data on white blood cell differentials; and the GLM (generalised linear model) statistical model results.

Supplemental Information 2 ARRIVE guidelines 2.0: author checklist.

Supplemental Information 3 MIQE Checklist.

We thank the Doctor of Veterinary Medicine Year 4 field ecology camp subject at the Melbourne Veterinary School in 2022 and Mikaeylah Davidson for collecting and processing the animals during the field study. We thank preclinical rotation students Huemay Semple, Emma Le Quesne, Hannah McIntyre, and Angus Tse for assisting with the sample processing and blood cell counts. We thank Abdul Jabbar and Charles Gauci for assistance identifying possible haemoparasites. Our collaborators Jessica Keem, David De Angelis, Nikolas Johnston, and Kevin Newman are compiling instances of ectoparasitic fly larvae found to be causing myiasis in frogs in Australia.

Additional Information and Declarations

Competing Interests

Author Contributions

Animal Ethics

Field Study Permissions

Data Availability

Laura Brannelly is an Academic Editor at PeerJ.

Tara Jadwani-Bungar conceived and designed the experiments, performed the experiments, authored or reviewed drafts of the article, and approved the final draft.

Nicholas P. Doidge performed the experiments, authored or reviewed drafts of the article, and approved the final draft.

Danielle K. Wallace performed the experiments, authored or reviewed drafts of the article, and approved the final draft.

Laura A. Brannelly conceived and designed the experiments, performed the experiments, analyzed the data, prepared figures and/or tables, authored or reviewed drafts of the article, and approved the final draft.

The following information was supplied relating to ethical approvals (i.e., approving body and any reference numbers):

The University of Melbourne Animal Ethics application number 20062.

Wildlife Act 1975 Research Authorisation for the Victorian Department of Environment Land Water and Planning permit number 1000984.

The following information was supplied relating to field study approvals (i.e., approving body and any reference numbers):

Wildlife Act 1975 Research Authorisation for the Victorian Department of Environment Land Water and Planning.

The following information was supplied regarding data availability:

The raw data is available in the Supplemental File.

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
