# Peer review of "Baseline haematological parameters in three common Australian frog species"

_PeerJ, doi:10.7717/peerj.17406_

## Round 0.1 · original submission · Major Revisions

Dear Dr. Brannelly,

Thank you for your submission. Reviewers found it a useful addition to the literature, but have made comments on a number of suggested changes to improve the manuscript. Reviewer 2 requested summary information of the sample populations, sex, species, size weight. This could be provided as a supplementary data table for by species. Regarding reviewer 2's comment on the data analysis-your analysis looks valid, I think if you explain to the reviewer that you used AIC to evaluate the best fit glm, that should suffice. Your microscopic photo images are excellent.

Sincerely,
Nicole L. Gottdenker

Reviewer 1 ·

Basic reporting

I thank the authors for their effort in explaining in introduction a clinically point of view. The professional English has been used throughout the text, and the overall structure of the manuscript is acceptable.
Anyway, to me the introduction needs some improvements and corrections as follow:

1- The three frog species necessarily need some explanation in taxonomic point of view.
2- The "et al." must be ended with a comma in the entire text.
3- The et al., sometimes id regular and sometimes is italic. Use the same instruction throughout the text.
4- Lines 43-45 needs one citation.
5- "and is present in Victoria, Australia." in Line 67, needs citation(s).

Experimental design

1- The questions and aims of the research are relevant and clearly stated.
2- The level of investigation, ethical and technical procedures is acceptable.

3- Regarding the methods some explanations must be presented:
3a- line 79: I think two questions should be answered here as adding some references; How species were identified? and What was the method for sex determination?
3b- Line 81: What preservative has been used?
3c- Lines 92-93: How long did it take from swabbing to storing at -20 degree?
3d- Line 111: Replace Biotechologies with Biotechnologies.
3e- Line 126: Have the amplified gene fragments been sequenced? and if yes, provide the accession number.
3f- Line 134: How long did it take to fixation and staining the blood smears?
3g- Lines 142-148: Is there any reference for the procedure used in these lines? or it is a modification to previous methods?
3h: Lines 176-177: Why Lit. paraewingi and not L. paraewingi? Is there any confusion for L. paraewingi in this manuscript with another scientific name? I recommend the authors to Follow Article 25 of ICZN.
3i- what is the significance levels for statistical analyses?

Validity of the findings

The results sections tries to present findings in relation to all questions and aims of the study, following the methods and procedures explained in the Method section. Nevertheless, some corrections and clarifications must be done in this section as follow:
Lines 213-214: "One C. signifera had an ectoparasite found in the blood smear"
Is it possible to find an ectoparasite in blood?

Line 224: How did you distinguish between adults and juveniles? base on which criteria? I recommend to explain it in the Material section.

Additional comments

Some corrections to figure legends have been highlighted which can be found in the attached text.

Annotated reviews are not available for download in order to protect the identity of reviewers who chose to remain anonymous.

Reviewer 2 ·

Basic reporting

Baseline haematological parameters in three common Australian frog species
(#90325)
This is an exploratory study of hematological variables in anuran species that
have experienced a decline in their populations. Nonetheless, a more comprehensive
description of the species regarding ecology, dietary habits, and risk of extinction in
the biome is necessary. The study's techniques require a more thorough explanation.
The text must be revised for accuracy and clarity. Results are unclear according to the
analysis methodology, and some aspects are missing from the discussion.
The high-quality images provide effective identification and classification of
cells in all studied species.

Experimental design

The experimental design is appropriate, although the model used (Akaike
Information Criterion - AIC) does not have a plausible explanation for its use. The
GLM model is feasible and commonly employed for this type of study.

Validity of the findings

Overall, the findings align with expectations for anuran species. Nevertheless,
the study methodology presents inconsistencies, especially in relation to the comparison
of leukocyte "proportionality" among species. The study does not mention the method, indicating that the authors developed it due to the limited number of samples per
specimen. Note that in very small samples of the total blood volume, the arrangement
and distribution of cells in the smear can be compromised, directly affecting the
ratio between leukocytes and erythrocytes. Thrombocytes may be affected in quantity
due to the quantity and quality of anticoagulant added to the blood. This significantly impairs
the quality of the sample, rendering a quantitative analysis of its elements. This
analysis, therefore, is invalid.
Data about the size and weight of the specimens (including gender) were
collected but not displayed. As such, the results and overall discussion are flawed and
do not align with the described conclusion.

Additional comments

General considerations for text and subitems

Line 75. Include here the basic information of the populations studied by species,
sample number, and sex. If possible, include the geographic coordinates (locations) of
the specimen sampled.

Line 77 - 79. ...frog collection and sampling were…. This statement may be removed;
there is a particular subtitle for ethics procedures.

Line 80. This is essential information for a biomass calculation for species; I suggest
Unlike the supplemental files, include these data in a table in the text body.

Line 82-83. This statement can be removed; it is a secondary information which is not
part of the study.

Line 128. Please consider replacing the subtitle: i.e., Blood and parasites analysis;
note: sampling is different from analysis.

Line 136. The author states the “differential blood cell count of leukocyte”…and
“differential cell count leukocyte.” Here, there is a conceptual error. The count should only
be used when cells are counted using a metric scale, such as a hematocytometer, for
erythrocytes and total leukocytes per blood volume. In this study, only a proportion of
cells were measured, which makes it difficult to make further comparisons between
individuals of the same species and between species. Differences in the balance of
fluids, circulating proteins, organic circulant salts, and collection stress can
drastically affect the sample. Unfortunately, controlling all of these
homeostatic attributes in these specimens is difficult, particularly during field collection and
with such a limited sample size.

Line 143-145. Here, provide one or two references for this methodology.

Line 174. Provide the acronym (AIC).

Line 200. I recommend including this information at the beginning of the methodology,
as it defines the experimental population, taking into account the sex of each individual.
All obtained results were based on this data, so they should not be included as results.

Line 207. The results of the parasitized specimens should be represented in a double
table for species and sex, informing the n for each sample. Line 211. Include a basic description of the species or group of parasites found.

Line 288-290. This is possible, but this hypothesis was not checked.

Line 300. Note that the conclusion is inconsistent with the initial objective of the study.

·

Basic reporting

The manuscript entitled “Baseline haematological parameters for three common Australian frog species” describes erythrocyte to leukocyte ratios and deferential leukocyte counts from free-ranging frogs. The captured frogs were noted for chytridiomycosis infection as well as presence of ectoparasites.
The research finding is interesting since haematological parameters present a useful method for determining the health status of animals. As leukocyte differentials change in response to stress, it is useful to evaluate whether the animal is under stress or not.

Experimental design

The research covered the following aspects:
To identify the different leukocyte cell types and create a visual guide
To determine the proportions of erythrocytes to leukocytes within blood smears from the three species
To determine leukocyte differential counts in the blood smears of field-collected individuals and assess association between parasites and differential counts.

Validity of the findings

No comments

Additional comments

Specific suggestions are stated below to improve the quality of the manuscript.
1. Page 7 line 44 <Davis, Maney & Maerz, 2008> to be replaced as Davis et al., 200
2. Page 7 line 46 <However, most free-ranging amphibian species we do not have the reference intervals necessary for accurate analysis of leukocytes (Davis et al. 2008, Forzán et al. 2017)> Sentence to be modified with addition of for <However, for most free-ranging amphibian species we do not have the reference intervals necessary for accurate analysis of leukocytes (Davis et al. 2008, Forzán et al. 2017).
3. Page 8 line 54 <(Ahmed, Reshi & Fazio, 2020)> to be replaced as Ahmed et al., 2020
4. Page 9 line 76 <in the Strathbogie Ranges, Victoria, Australia (n = 126)> Geographic location to be included for clarity of study area.
5. Page 9 line 82 <Keem, D De Angelis, N Johnston, and K Newman> affiliation of collaborators to be mentioned.
6. Are the ectoparasites only larvae of flies or any other species?
7. Page 9 line 84 <All frogs were swabbed for B. dendrobatidis following standard methods (see below)> Reference for the method to be mentioned
see below to be replaced with as stated below.
8. Page 12 line 154 <we were unable to perform molecular analysis on the sample to identify any haemoparasites that might have been present.
Page 11 line 138 and 139 <detect the presence of any haemoparasites.>
Both statements are contrasting
9. Fig. 1a Is the parasite held with the forceps? If so, in Fig. 1b the magnified parasite does not match with Fig. 1a
10. Fig 2. Legend last line <the scale bar indicates 20mm> but in Fig. 2 the scale bar is showing µm
11. Authors should have shown in a separate figure all 4 ectoparasite fly larvae attached to the frog for clarity of readers.
12. Fig. 3 legend < The scale bar indicates 20mm> replace with 20 µm
13. Page 14, line 212< very few mature erythrocytes (Fig 2).> very few may be represented in terms of number as compared to the normal frog of the same species.
14. Page 16 line 240<The impacts of parasites on cell differentials> which parasite to be mentioned, ie., ectoparasites (larvae of fly) or fungal infection with chytridiomycosis
15. Page 17 line 269 <because none of the Lim. dumerilii sampled> samples or sampled
16. Page 17 line 279 <heavily infected with an ectoparasite> please specify the number of ectoparasites (maximum number stated is 4 in the text)

It is further suggested to improve the English language of the manuscript with the help of any professional editing service.

The manuscript describes for the first-time haematological parameters for three common Australian frog species with clear species differences in the visual
appearance of blood cells. Each species has a unique leukocyte profile. Due to small sample size, effect of parasites on blood cell profile has not been correlated which may be addressed in future studies.

---

## Round 0.2 · Minor Revisions

Dear Authors,

I believe that all of the reviewers' comments have been well-addressed. I have only one very minor comment. Can you provide summary tables of glm results in a supplementary table? Unless I have missed this, they are not in the supplementary data submitted. Sorry I may have missed them.

---

## Round 0.3 · Minor Revisions

Thank you for your revision. Reviewer 3 has added a handful of minor revisions (about a list of 5 suggestions), including showing all 4 ectoparasites attached to frogs in a separate figure for reader clarity, and the addition of some references and correction of references.

Reviewer 1 ·

Basic reporting

The article meet the Basic reporting standards.

Experimental design

The article has met all the standards of experimental design.

Validity of the findings

The article has met the standards of this section.

Additional comments

I wish to thank the authors for conducting this original research which can be used as a potential source to compare future works with.

I also thank the authors for considering and correcting the suggestions and comments of the referee. So, I have decided to accept this article.

·

Basic reporting

Major issues have been resolved in the revised manuscript.

Experimental design

Mentioned in the 1st review

Validity of the findings

mentioned in the 1st review

Additional comments

Majority of issues have been resolved. However, five points mentioned below as per the 1st draft could not be traced because of change in page and line number of the 2nd draft.
6. Are the ectoparasites only larvae of flies or any other species?
8. Page 12 line 154 <we were unable to perform molecular analysis on the sample to identify any haemoparasites that might have been present.
Page 11 line 138 and 139 <detect the presence of any haemoparasites.>
Both statements are contrasting
11. Authors should have shown in a separate figure all 4 ectoparasite fly larvae attached to the frog for clarity of readers.
13. Page 14, line 212< very few mature erythrocytes (Fig 2).> very few may be represented in terms of number as compared to the normal frog of the same species.
16. Page 17 line 279 <heavily infected with an ectoparasite> please specify the number of ectoparasites (maximum number stated is 4 in the text)

Below are two corrections to be done on the 2nd draft
56 intrinsic and extrinsic factors (Ahmed, Reshi & Fazio, 2020). Edit the reference <Ahmed et al., 2020>

283 similar to mammals and birds (Davis, Maney & Maerz, 2008). Edit the reference <Davis et al., 2008>

Authors are advised to cite some haematological reports on anurans from other parts of the world in the manuscript to have a comparative account.

Some references are cited below

Bana Bihari Mahapatra B B, Das M, Dutta S K and Mahapatra PK (2012) Hematology of Indian rhacophorid tree frog Polypedates maculatus Gray, 1833 (Anura: Rhacophoridae) Comparative Clinical Pathology, 21:453-460

Hota J, Das M and Mahapatra PK (2013) Blood cell profile of the developing tadpoles and adults of the ornate frog, Microhyla ornata(Anura: Microhylidae). International Journal of Zoology, http://dx.doi.org/10.1155/2013/716183.

Das M and Mahapatra PK (2015) Blood cell profile of the Indian Tree Frog Polypedates maculatus (Gray, 1830), during larval development until metamorphosis (anura: rhacophoridae) Herpetozoa 27(3/4):123-135.

Das M and Mahapatra PK (2016) Blood cell profile of the tadpoles of Chirixalus simus (Anura: Rhacophoridae) Russian Journal Herpetology. 23 (2) 83-92.

---

## Round 0.4 · accepted · Accept

All reviewers' comments were well-addressed and I think the manuscript is ready for publication.